# Re-Developing the Adversity Response Profile for Chinese University Students

**DOI:** 10.3390/ijerph19116389

**Published:** 2022-05-24

**Authors:** Xiang Wang, Zi Yan, Yichao Huang, Anqi Tang, Junjun Chen

**Affiliations:** 1Department of Curriculum and Instruction, The Education University of Hong Kong, Tai Po, Hong Kong, China; wxiang@eduhk.hk (X.W.); s1139510@s.eduhk.hk (A.T.); 2Department of Humanities and Foreign Languages, China Jiliang University, Hangzhou 310018, China; freda_abc@163.com; 3International Affairs Office, Huangshan University, Huangshan City 245000, China; 4Department of Education Policy and Leadership, The Education University of Hong Kong, Tai Po, Hong Kong, China; jjchen@eduhk.hk

**Keywords:** adversity response profile, adversity quotient, scale re-development, Rasch measurement

## Abstract

Adversity response is fundamental to dealing with adversity. This paper reports the re-development and subsequent psychometric evaluation of the Adversity Response Profile for Chinese University Students (ARP-CUS). The data were collected from a Chinese university student sample (*n* = 474). Factor analysis and Rasch analysis were used to examine the psychometric properties of the ARP-CUS. Exploratory factor analysis revealed a six-factor model; then confirmatory factor analysis supported a five-factor solution. Rasch analysis provided further evidence of the psychometric quality of the instrument in terms of dimensionality, rating scale effectiveness, and item fit statistics for those six dimensions. The final version of the ARP-CUS contains 24 items across five subscales for assessing students’ responses to adversity, including control, attribution, reach, endurance, and transcendence. Overall, ARP-CUS demonstrates satisfactory psychometric properties for quantifying the adversity quotient of Chinese university students.

## 1. Introduction

The adversity quotient (AQ) is an assessment of “how well a person can withstand adversity and his/her ability to surmount any crisis” [1]. The importance of the adversity response is such that the AQ has been claimed to be comparable with other personal quotients (e.g., Intellectual Quotient; Emotional Quotient; Spiritual Quotient) [2,3], and has attracted increasing interest among researchers and practitioners across a range of cultures [4,5,6,7]. It is also an appropriate but understudied area for studies of Chinese university students. China has a tertiary population of 41,830,000 students in 2738 universities [8]. Past studies have shown that Chinese university students suffer from a variety of hardships and adversities, both at university, and in society more generally, because of low self-control and lack of self-awareness [6,9]. This suggests that university students’ AQ should be a fruitful area for further research in China. One significant hurdle faced by researchers in this field is a lack of appropriate instruments for assessing students’ AQ [10]. Most response to adversity studies in the Chinese context has used the adversity quotient scale (AQS) [11], a translated version of the original Adversity Response Profile (ARP) developed by Stoltz [12]. In this context, the translated scale has two potential weaknesses. Firstly, as the AQS originates from Western culture, it might not cover aspects of adversity response that are unique to Eastern culture. For example, the Malaysian Youth Adversity Quotient Instrument (MY-AQi) emphasized elements appropriate in the Malaysian context (e.g., political and technological elements), although its development was also based on Stoltz’s AQ theory [13]. In terms of the Chinese population, Confucianism and Taoism are two predominant elements that influence Chinese culture, affecting how the Chinese interpret adversity situations and respond to them [14]. Cheng reported that a suffering experience can be perceived as a disaster but can also be interpreted as an opportunity for harmony and self-transcendence by the Chinese [15]. Regarding Chinese students, the ethic of self-cultivation and self-transcendence play a crucial role when they assess and handle college stressors [16]. Secondly, the AQS was developed initially for individuals in workplace environments. As adversity response is context-dependent [1] and the adversities faced by students and working persons are potentially different, some items in the AQS might not be directly applicable to learning environments (see more details in Section 1.2). To address these methodological gaps, the current study aims to re-develop an instrument based on the ARP and AQS to assess the adversity response of Chinese university students. The new tool would extend the application of the original AQ scales to be: (1) more culturally sensitive, and (2) more appropriate for students in learning contexts.

### 1.1. Adversity Quotient

AQ is an indicator assessing how one withstands, overcomes or deals with the adversities of one’s life [1,12]. The AQ concept draws insights from several branches of human and social sciences, including cognitive psychology (control and mastery of one’s life), psychoneuroimmunology (immune function), and neurophysiology (the science of the brain). In Stoltz’s framework, adversity response consists of five major sub-constructs, named CO_2_RE, representing Control, Origin, Ownership, Reach, and Endurance. Control refers to the sense of control toward adversity situations and individual responses. Origin refers to the source of the problem. Ownership concerns taking responsibility in adversity situations. Reach refers to how far adversity would influence one’s life. Endurance refers to how long the hardship and its causes might endure [12].

In parallel with theoretical arguments, past empirical studies have indicated that AQ has close links with motivation, mental stress, perseverance, learning, and response to changes [7,17,18]. In the field of education, AQ plays a crucial role, being correlated positively with achievement motivation and academic performance [9,19,20,21,22]. For instance, the achievement motivation of college nursing students could be predicted by their AQ [9]. AQ correlated positively with motivation to succeed and achievement motivation (*r* = 0.34, *p* < 0.01), and negatively with motivation to avoid failure (*r* = −0.37, *p* < 0.01) for financially poor college students [22]. Students with higher levels of AQ performed better in learning English [21] and mathematics [23], as well as leadership skills [24].

### 1.2. Instruments Assessing Adversity Quotient

The Adversity Response Profile (ARP) was developed by Stoltz [1]. This 40-item instrument assesses five AQ dimensions: C (control), O_2_ (origin and ownership), R (reach), and E (endurance). It also exists as a 20-item short form with just 20 adversity scenarios [1]. Li and Chen translated the ARP into Chinese and renamed it the Adversity Quotient Scale (AQS). They examined its reliability and validity with a data set from 606 Chinese students across primary school, secondary school, and university. Cronbach’s alpha values ranged from 0.71 to 0.81 for the subscales, and test-retest reliability values ranged from 0.73 to 0.79 for the five sub-constructs. However, some indicators of the fit statistics (GFI (the goodness of fit index), AGFI (the adjusted goodness fit index), and NNFI (non-normed fit Index)) in CFA (confirmatory factor analysis) were not satisfactory (χ^2^ (590) = 1848.21, GFI = 0.85, AGFI = 0.84, NNFI = 0.75), indicating this tool needed further improvement for use with Chinese students [11].

The unsatisfactory CFA results for the Chinese AQS might be attributed to two conceptual limitations associated with the scale. Firstly, it does not consider the influence of culture, which significantly impacts the types of adversities experienced, the resources available to deal with adversities, the perception of adversities, coping strategies, and adaptation outcomes [25,26]. In Western cultures, cultivating rigidity and avoiding helplessness are essential elements in responding to adversity by emphasizing control of, and then improving the adversity situation [1,12]. In contrast, accepting adversity is a unique but essential value in Eastern cultures [27]. People in Chinese society value harmonious social relationships and usually consider adversity as “suffering” determined by fate [26]. Generally, individuals either accept adversity honestly from the perspective of Confucianism [28] or accept adversity with a peaceful mind from the philosophy of Taoism [29]. Because of the belief that people need to repay the debts of their previous life, acceptance of adversities might be regarded as a necessity and self-cultivation. Thus, we propose the inclusion of a new AQ dimension, i.e., “transcendence” for research in Chinese cultural contexts. Transcendence is a sense of positive acceptance of adversity situations. It refers to affinity towards, rather than alienation from, situations of adversity; advocating for taking advantage of adversity actively instead of running away. In other words, it refers to beliefs of self-acceptance, i.e., various suffering has already been determined by fate, and those tough times are beneficial to life. Past studies have demonstrated similar cultural differences in related variables, such as coping. Some studies identified transcendence as a unique coping mechanism for athletes in Eastern culture. This is not surprising as coping styles are argued to be closely related to AQ [30,31]; individuals with high AQ would choose positive coping styles, while individuals with low AQ would adopt negative coping styles [12].

Secondly, the original ARP items were developed for workplace environments and, therefore, might not be relevant to the situation of learning: some items (e.g., To hold your position, you must be repositioned; You were demoted or punished) are not suitable in student contexts.

### 1.3. Current Study

In response to the abovementioned methodological gaps, the present study aims to (1) redevelop the ARP specifically for Chinese university students; and (2) examine its psychometric characteristics. We redeveloped the ARP by adding a Chinese culture-based dimension of AQ, transcendence, a unique psychological mechanism of self-acceptance when facing adversity that is rooted deeply in Eastern cultures. We also revised some items of the ARP to fit university learning contexts better.

## 2. Materials and Methods

### 2.1. Participants

A convenience sample of 498 students from two Chinese universities, one in Zhejiang and the other in Anhui provinces, were invited to participate in the study. Valid responses were collected from 474 students (232, 48.95% from Zhejiang). The mean age was 20.45 years (range = 18–31 years, *SD* = 1.47 years), and the majority were female (65.2%, see Table 1 for details).

### 2.2. Instruments

The Adversity Response Profile for Chinese University Students (ARP-CUS) was developed for this study. All items in the original ARP were reviewed and revised by the research team to fit the context of the Chinese university students. The items in the new dimension transcendence were developed based on a literature review, available instruments assessing relevant variables, such as Coping Scale for Chinese Athletes (CSCA) [32], and a focus group interview with 12 university students. Those procedures resulted in a 25-item scale with five items in each of the five subscales: control, O_2_ (origin and ownership), reach, endurance, and transcendence. All items were developed and revised in Mandarin Chinese.

These 25 items were reviewed by a panel of five experts from the fields of psychology and/or educational assessment in terms of content validity and readability. The content validity ratios (CVRs) were computed based on experts’ responses to the question “Is this item ‘essential’, ‘useful, but not essential’, or ‘not necessary’ to assess student adversity quotient?” Items with positive CVRs (i.e., the number of ‘essential’ responses is three or more) were retained [33,34]. In addition, experts were asked to sort the items into dimensions. Items were retained when at least four experts correctly classified the dimension [33,34]. In addition, we invited experts to comment on the readability of the items and made amendments in light of their feedback. Consequently, five items with negative CVRs and/or incorrect dimension sorting results were discussed and revised. A resultant 25-item survey was generated under five subscales: 4 items were literal translations of the original ARP items, 16 were modified, and 5 were newly developed. A standard five-point Likert-type response scale was adopted for all items. The item development is coded in Table 2: O = original item; M = modified item; N = new item. The English version of the instrument is provided in Appendix A. The items were translated by two bilingual (English/Chinese) academics specializing in educational psychology, following the standard forward- and backward-translation procedure [35]. It should be noted that the English version has not been empirically validated. The Chinese version is available upon request.

Short Form of achievement motive scale (AMS-SF). McClelland argued that people with high achievement motivation were more likely to be ambitious people who would try to fight against adversity [36]. In addition, Lin and Chen indicated that individuals with a high level of adversity quotient were more likely not to be afraid of failure [22]. Thus, we used achievement motivation as a criterion indicator and hypothesized that the ARP-CUS scores would correlate positively with the motive to achieve success and negatively with the motive to avoid failure. The achievement motive scale (AMS) is a 30-item instrument assessing the motive to achieve success (Ms) and the motive to avoid failure (Mf) [37]. Tang and Lu developed a short form of the achievement motive scale (AMS-SF) for Chinese students. It consisted of six items for each dimension and showed good reliability (Cronbach’s alpha for Ms and Mf were 0.81 and 0.85, respectively) [38]. We hypothesized that the ARP-CUS scores would correlate positively with MS, the motive to achieve success, and negatively with Mf, the motive to avoid failure.

### 2.3. Procedure

Data were collected anonymously through an online survey, which students completed in less than 15-min. Informed consent was obtained, and participants were informed that they had the right to withdraw from the study at any time without any negative consequences. Therefore, the willingness of participation was recorded by completion of the online questionnaire, and those who were not taken part were thanked and the survey terminated. The study was approved by the Human Research Ethics Committee of the first author’s affiliated university.

### 2.4. Data Analysis

The psychometric properties of ARP-CUS were examined using two analytical approaches, factor analysis and Rasch analysis, to provide more comprehensive evidence. The data for factor analysis were divided randomly into two subsamples. Subsample 1 was used for exploratory factor analysis (EFA) with SPSS (version 27.0, IBM Corp, Armonk, NY, USA) to identify the instrument’s factor structure, and subsample 2 was used for confirmatory factor analysis (CFA) with M-plus (version 8.3, Muthén & Muthén, Los Angeles, CA, USA) to verify that factor structure. A sample size of 200 is sufficient in most cases of ordinary factor analysis that involve no more than 40 items [39]. DeVellis recommended a person-item ratio between 5:1 to 10:1 for factor analysis [40]. We have 25 items in the instrument and more than 200 cases for EFA (232) and CFA (242) in this study. Therefore, the sample size was sufficient for our analytical purposes. In addition, further details of psychometric properties of ARP-CUS were provided by Rasch analysis applied to the whole data set. Finally, the Cronbach’s alpha coefficients for each ARP-CUS subscale, and correlation coefficients between the ARP-CUS and the AMS-SF subscales were computed.

EFA was first conducted with the principal axis factoring method and Promax rotation analysis [41]. Factor structure was determined by eigenvalue text, scree plot, and interpretability of the rotated factors. In addition, deletion criteria were: factor loading less than 0.3 and cross-loading above 0.3 for an item. After the factor structure was identified, CFA was conducted with maximum likelihood estimation [42,43]. Model fit was evaluated using multiple fit indices, including the Tucker-Lewis index (TLI), the comparative fit index (CFI), the standardized root mean square residual (SRMR), and the root-mean-square error of approximation (RMSEA) [44]. Acceptable values of greater than 0.90 were adopted for TLI and CFI, and less than 0.08 for SRMR and RMSEA [45,46].

Rasch analysis has been widely applied and advocated in social science research, especially for examining instrument quality [47]. Empirical studies have demonstrated that using factor analysis and Rasch analysis can evaluate instrument quality more comprehensively [48,49,50]. As, theoretically, AQ is a multidimensional construct, we applied a multidimensional Rasch model using ConQuest 2.0 [51,52]. Four indicators were used to check the instrument quality, including *Item fit statistics* (i.e., Infit MNSQ and Outfit MNSQ), *Differential item functioning* (DIF), *floor/ceiling effects*, and *Rasch reliability*.

We examined five aspects of validity from Messick’s validity framework [53]. Content validity was ensured by the theory-driven procedure of scale development and expert review. Substantive validity was investigated by the data-to-model fit statistics in Rasch analysis. Structural validity was examined by EFA and CFA, and generalizability validity was examined by DIF analyses. External validity was tested by the correlation between ARP-CUS and achievement motivation. However, evidence for consequential validity was beyond the scope of this developmental study.

## 3. Results

### 3.1. Exploratory Factor Analysis (EFA)

The results showed that values for skewness and kurtosis were below the thresholds (3.0 and 8.0), indicating that the data from subsample 1 were close to normally distributed. The values of Kaiser-Meyer-Olkin of 0.79, and Bartlett’s test of sphericity, χ^2^ (276) = 1576.49, *p* < 0.001, indicated that the data were appropriate for factor analysis. Six factors were extracted with eigenvalues higher than one, which explained 56.72% of the total variance cumulatively. The scree test also supported the six-factor solution. Twenty-four items loaded on the six factors as specified by the theoretical model, with factor loadings greater than 0.30 and without competing cross-loading [54]. The only exception was item T1 (Your efforts were not recognized in group work. You believe that it is not necessary to care about it but just let it go). This item was designed as an item for the transcendence dimension, but it did not clearly load on any of the six factors. Therefore, this item was removed and the EFA was repeated (see Table 2 for the results).

### 3.2. Confirmatory Factory Analysis

CFA was conducted on a subsample (*n* = 242) to check the factor structure. In addition to the six-factor first-order model identified by EFA (model 1, see Figure 1), another two models (model 2, see Figure 2 and model 3, see Figure 3), which were consistent with the original theoretical model of Stoltz in 1997, were also examined. In model 2, O_2_ was a combination of the origin and ownership factors [12]. In model 3, origin and ownership factors contributed to a second-order factor, namely, attribution [11]. It can be seen from Table 3 that model 3 has better fit statistics than model 1 and model 2, implying the factor solution in model 3 accounted for the data better than model 1 and model 2. In addition, the standardized factor loading of each item was acceptable for all.

### 3.3. Rasch Analysis

Rasch analysis was applied to the 24 items confirmed by factor analysis on the full dataset (*n* = 474). Although a five-point response scale was applied to all items, the descriptors of the scales varied across subscales. Thus, we employed a partial-credit Rasch model [55]. The results showed that all items had a satisfactory fit to the Rasch model, i.e., Infit and Outfit MNSQ falling within the accepted range of 0.75 to 1.33 [56]. This finding indicates that all items measure the latent trait as theoretically expected. DIF was examined by calculating the difference in item difficulty across groups after controlling the latent trait levels. As suggested by Wang et al., a difference of item difficulty equal to or larger than 0.5 logits implies substantive DIF [57]. The analysis did not identify any item showing substantial DIF across gender, university, or grade, indicating the instrument works invariantly across different groups of respondents. Floor and ceiling effects were also checked. The results showed that the percentage of respondents at the highest and lowest ends of the six subscales was less than 0.5%, indicating there were no floor or ceiling effects [58].

### 3.4. Criterion-Related validity

The AMS-SF showed satisfactory reliabilities with the sample in the current study (the Cronbach’s alpha for Ms and Mf were 0.85 and 0.86, respectively). The results indicated that the ARP-CUS measures had a significantly positive correlation with Ms, the motive to achieve success (*r* = 0.31, *p* < 0.01) and a negative correlation with Mf, the motive to avoid failure (*r* = −0.14, *p* < 0.01) (see Table 4 for details). These results were consistent with our hypotheses and the results of the previous study [9].

### 3.5. Reliability

Both Cronbach’s alpha coefficients and Rasch reliabilities were computed for each ARP-CUS subscale, showing acceptable reliability on the total sample (*n* = 474). Cronbach’s alpha was 0.71 for control, 0.64 for origin, 0.74 for ownership, 0.80 for reach, 0.79 for endurance, and 0.72 for transcendence, respectively. The Rasch reliabilities for the six subscales were 0.72, 0.72, 0.62, 0.73, 0.74, and 0.70, respectively. The reliabilities for the subscales of origin and ownership were relatively low, but still acceptable given that these two subscales had only three and two items, respectively.

## 4. Discussion

This study aimed to redevelop and validate an instrument for assessing Chinese university students’ adversity quotient. The final ARP-CUS contains 24 items across six dimensions: control, origin, ownership, reach, endurance, and transcendence. The results showed good psychometric properties of the developed scale, with both factor analysis and Rasch analysis supporting the six-factor higher-order model with a new dimension, ‘transcendence.’

Culture can be a crucial factor impacting the efficacy of measurement tools [59]. An instrument working well in one culture might not work well for people from a different culture. The original ARP was developed in Western culture [12]; therefore, it might be challenging to accurately evaluate the ability of the Chinese individual to respond to adversity. The unsatisfactory psychometric properties of the Chinese version of ARP attest to this challenge [11]. The new ARP-CUS dimension, transcendence, provides a possible solution to tackle this problem. We integrate traditional Chinese philosophies, Confucianism and Taoism with the latest Western theorization of adversity quotient [1] to discover aspects of adversity quotient that are indigenous to the Chinese culture as well as applicable to the Eastern culture. This finding also has the potential to enrich the AQ theory by echoing the importance of transcendence highlighted in previous studies. For instance, in addition to the three main methods for coping with stress in Western culture, problem-focused coping, emotion-focused coping, and avoidance coping [60], Yoo identified another unique way to handle stress, namely transcendence coping, among Koreans [31]. Later, transcendence coping was also identified among the Chinese [32].

We then examined the relationship between ARP-CUS and achievement motivation. The results showed that the ARP-CUS score had a significantly positive correlation with achievement motivation. Individuals with a high ARP-CUS score perceived more control and less enduring damage from difficulties and challenges. This finding was in line with past studies and highlighted that perception control and endurance correlated significantly with Ms and Mf [9,22]. Interestingly, the transcendence dimension had a positive correlation only with Ms (*r* = 0.32, *p* < 0.001), but not with Mf. This finding is different from the theoretical expectation. According to the self-worth theory, the highest human priority is the exploration of self-acceptance, and this need may promote both an orientation to approach success and avoid failure [14,61]. Since self-acceptance is a crucial component of transcendence, it is reasonable to expect that transcendence has a positive relationship with motivation to approach success and avoid failure. A possible explanation is that students with a higher score of transcendence could be more likely to improve themselves by overcoming adversity actively, and thereby be more confident of achieving success. At the same time, their self-acceptance tendency might not be consistent with the motive to avoid failure. Nevertheless, as this is the first validation of the transcendence dimension, future studies can test these relationships in different samples.

The ARP-CUS can facilitate future research on AQ in China and the regions sharing similar Eastern cultures, such as East and Southeast Asia. With the ARP-CUS, researchers and practitioners can depict more comprehensive AQ profiles for students. Such profiles could be crucial for teaching and learning, given that there are close associations between AQ and various learning outcomes, such as mathematics achievement [23], logical thinking ability [62], and creative mathematical reasoning ability [5]. Moreover, the AQ profile provided by the ARP-CUS might help in the design of appropriate intervention or counseling programs to increase students’ AQ more efficiently by putting more emphasis on the areas that need more support.

Although the present study provided initial evidence for the Simplified Chinese ARP-CUS, several limitations should be noted. Firstly, the number of items in the origin and ownership dimensions is three and two. Future studies might consider developing more items for these two dimensions if they are deemed essential in a particular cultural or research context. Secondly, the sample was limited to students from southeast China, and most of them were female. It will be helpful to examine the scale’s psychometric properties using samples from other Eastern countries and to balance more evenly the number of males and females. Thirdly, this study did not examine the consequential validity of the scale. Future studies might investigate what particular beneficial outcomes could be obtained with the use of the scale.

## 5. Conclusions

In conclusion, the development of the ARP-CUS provides a psychometrically sound and valuable tool for assessing the adversity quotient in Chinese university students. The information provided by this scale can contribute to a better understanding of Chinese university students’ ability to respond to adversity and, therefore, inform teachers and practitioners about how students’ adversity quotient might be enhanced.

## Figures and Tables

**Figure 1 ijerph-19-06389-f001:**
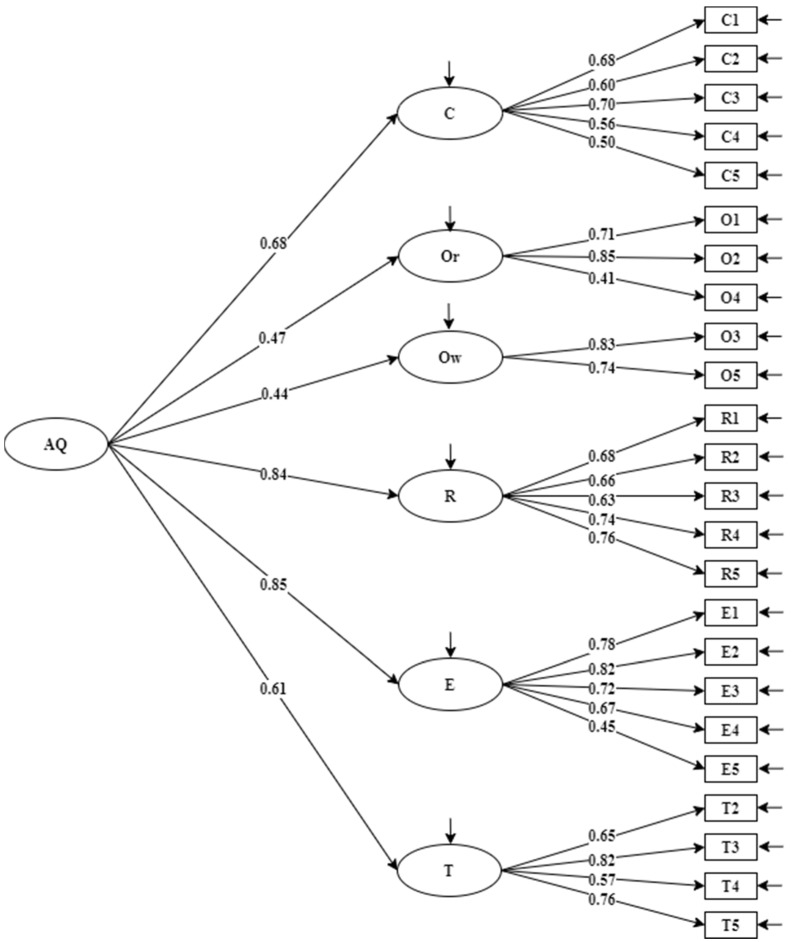
Model 1 with standardized factor loadings on control (C), ownership (Ow), origin (Or), reach (R), endurance (E), transcendence (T), and adversity quotient (AQ) (*n* = 242).

**Figure 2 ijerph-19-06389-f002:**
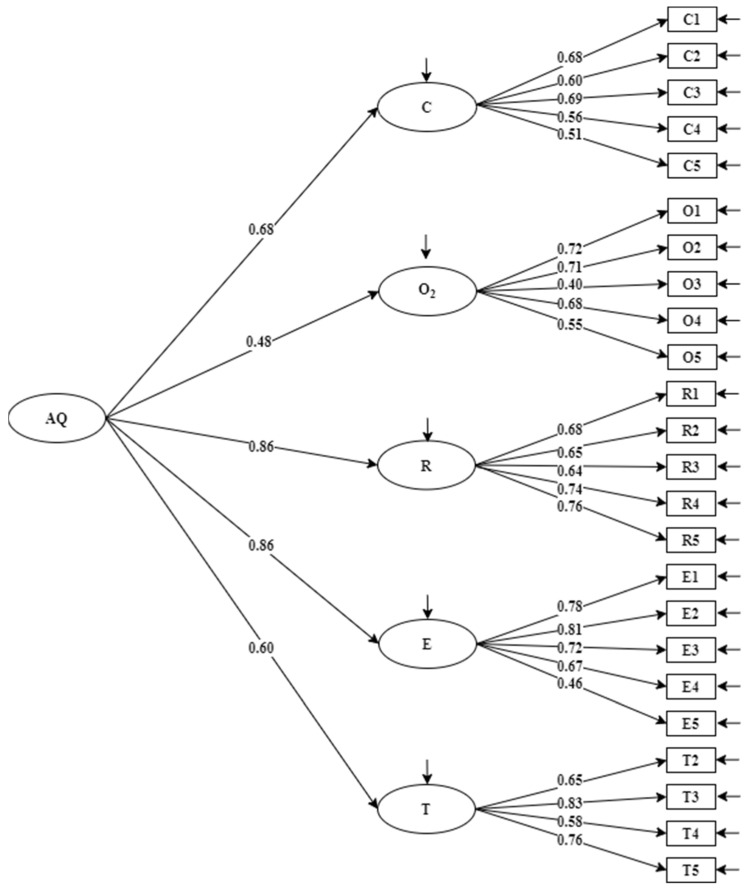
Model 2 with standardized factor loadings on control (C), ownership and origin (O_2_), reach (R), endurance (E), transcendence (T), and adversity quotient (AQ) (*n* = 242).

**Figure 3 ijerph-19-06389-f003:**
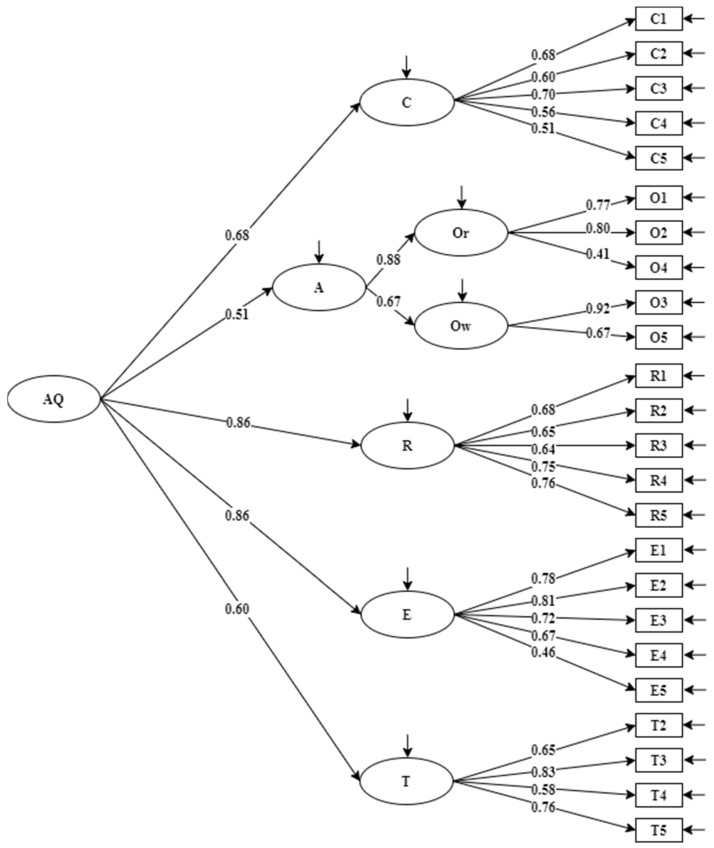
Model 3 with standardized factor loadings on control (C) and attribution (A), which consists of ownership (Ow), origin (Or), reach (R), endurance (E), transcendence (T), and adversity quotient (AQ) (*n* = 242).

**Table 1 ijerph-19-06389-t001:** Demographics of participants.

	Frequency	Percentage
Year of study		
First	141	29.8
Second	85	17.9
Third	153	32.3
Fourth	95	20
Gender		
Male	165	34.8
Female	309	65.2
Source		
City	151	31.9
Urban	323	68.1
*n*	474	100

**Table 2 ijerph-19-06389-t002:** Factor loadings for Items on sample 1 (*n* = 232).

Item	Status	EFA Factor Loadings					
		C	Or	Ow	R	E	T
C1: You suffer an academic setback.	M	**0.46**	0.05	0.12	0.23	−0.04	0.00
C2: People don’t like your idea during a discussion.	M	**0.47**	−0.03	−0.05	0.10	0.05	0.00
C3: Your personal and study obligations are out of balance.	M	**0.71**	0.08	−0.01	0.05	−0.08	−0.02
C4: You have a conflict with your family.	M	**0.77**	−0.16	0.07	−0.20	0.07	−0.01
C5: Your computer crashed for the third time, wasting your time.	O	**0.36**	−0.06	−0.05	0.00	0.08	0.06
O1: You are overlooked for the opportunity of being given an excellent person award.	M	0.07	**0.50**	0.30	−0.05	−0.08	0.06
O2: Someone you respect ignores your attempt to discuss an important issue.	O	0.07	**0.72**	0.07	−0.12	0.08	−0.01
O4: One of your important friends did not show up on your birthday.	M	−0.23	**0.53**	0.09	−0.02	0.03	−0.08
O3: You fail to complete the work arranged by the teacher.	M	0.15	0.12	**0.66**	0.02	−0.01	−0.08
O5: You fail a specific course.	N	−0.08	0.16	**0.76**	0.08	0.02	0.07
R1: You are criticized for a subject assignment.	N	0.00	−0.14	0.25	**0.62**	0.10	−0.06
R2: The important activity you are taking on gets canceled.	M	−0.03	−0.10	0.07	**0.69**	−0.03	0.04
R3: You go through a significant number of bad patches in one day.	M	−0.06	0.21	−0.24	**0.65**	0.04	−0.02
R4: You miss an important appointment.	O	0.04	0.04	−0.04	**0.68**	−0.04	0.10
R5: Your teacher adamantly disagrees with your idea.	M	0.04	−0.18	0.07	**0.64**	0.07	−0.05
E1: You accidentally delete an important message.	M	0.05	−0.03	0.06	0.10	**0.53**	−0.09
E2: You argue with someone and develop negative emotions.	M	0.02	0.08	−0.04	0.05	**0.76**	−0.02
E3: You leave some messages for a friend, but without any reply.	M	−0.06	0.00	0.01	−0.07	**0.79**	0.06
E4: You missed a flight or a train when you were traveling.	M	0.03	0.00	0.01	0.09	**0.49**	0.02
E5: You lost something important to you.	O	0.12	0.13	−0.24	0.05	**0.32**	0.02
T2: You believe that it is beneficial to tactically compromise when arguing with a friend.	N	0.04	−0.08	0.10	−0.15	0.26	**0.39**
T3: Even though you put in lots of effort, you still failed the exam. You believe this is a necessary step for success.	N	−0.15	−0.05	0.11	0.04	0.01	**0.84**
T4: You have no idea about how to complete work assigned by a teacher. You believe that it is not necessary to worry about it; it can be addressed eventually.	N	0.11	−0.03	−0.16	0.02	−0.09	**0.47**
T5: You go through lots of bad patches during a period. You believe this is a good chance to strengthen your will.	N	0.17	0.11	−0.09	0.07	−0.04	**0.48**
Eigenvalues		1.72	1.30	1.40	5.15	2.57	1.47
% variance explained		7.17	5.42	5.81	21.45	10.72	6.14

Note: The values shown in bold are the rotated factor loading of the derived six factors. Extraction method: principal axis factoring. Rotation method: Promax with Kaiser Normalization. Status: O = original item; M = modified item; N = new item. C = control; Or = origin; Ow = ownership; R = reach; E = endurance; T = transcendence.

**Table 3 ijerph-19-06389-t003:** CFA goodness-of-fit indices for the models on Sample 2 (*n* = 242).

	TLI	CFI	SRMR	RMSEA
Model 1	0.897	0.884	0.070	0.055
Model 2	0.872	0.886	0.065	0.058
Model 3	0.903	0.914	0.063	0.050

Note: TLI = Tucker-Lewis index; CFI = comparative fit index; SRMR = standardized root mean square residual; RMSEA = root mean square error of approximation. Model 1: six-factor model; Model 2: five-factor model grouping the factors Or (origin) and Ow (ownership) in one factor; Model 3: Or (origin) and Ow (ownership) factors contributed to a second-order factor A (attribution).

**Table 4 ijerph-19-06389-t004:** Model Means of Rasch-calibrated measures, standard deviations, and correlations between each subscale and the achievement motivation scale.

	Mean	*SD*	1	2	3	4	5	6	7	8
1. MS	−0.04	2.30	-							
2. MF	1.07	2.20	−0.122 **	-						
3. Control	0.86	1.06	0.207 **	−0.242 **	-					
4. Origin	1.14	1.18	0.165 **	0.004	0.261 **	-				
5. Ownership	1.22	1.33	0.092 *	0.042	0.054	0.535 **	-			
6. Reach	0.70	1.40	0.127 **	−0.131 **	0.429 **	0.092 *	0.069	-		
7. Endurance	0.45	1.29	0.272 **	−0.176 **	0.421 **	0.159 **	0.054	0.584 **	-	
8. Transcendence	0.51	1.15	0.330 **	−0.016	0.337 **	0.189 **	0.102 *	0.364 **	0.404 **	-

Note: * *p* < 0.5, ** *p* < 0.01.

## Data Availability

The datasets generated during and/or analyzed during the current study are available from the corresponding author on reasonable request.

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
