# Peer review of "Re-Developing the Adversity Response Profile for Chinese University Students"

_ijerph, 2022, doi:10.3390/ijerph19116389_

Round 1

Reviewer 1 Report

The manuscript is a pleasure to read. The conciseness is something to be commended.

  1. Because the identified research gap includes the effects of culture on the measurement of AR, it would be desirable to report what had been found regarding AR in other cultures.
  2. It is not clear how many items were generated for Transcendence. Were there exactly 5 items? Normally, it is advisable that the pool of items exceeds the target number of items. 
  3. For the Rasch analysis, it would be good to report the floor/ceiling effect.
  4. The results of the Multidimensional Rasch Model analysis could be elaborated. The infit&outfit indices seem to be too brief to convince the readers that the ARP-CUS has 5 dimensions.
  5. Additionally, since the rating scales use different anchors, it should be worth mentioning that a Partial Credit Model was used.  

Author Response

Dear Reviewer,

Thanks for your comments.

Best

WX

Reviewer 2 Report

This is a clearly written paper, which I have enjoyed reading.  It reports the re-development and subsequent psychometric evaluation of the Adversity Response Profile for Chinese University Students (ARP-CUS). I can see that the data were collected from a Chinese university student sample (n = 474). This is a good sample for validating the redeveloped scale. Factor analysis and Rasch analysis were used to examine the psychometric properties of the ARP-CUS. Exploratory factor analysis revealed a 6-factor model, then confirmatory factor analysis supported a 5-factor solution. Rasch analysis provided further evidence of the psychometric quality of the instrument in terms of dimensionality, rating scale effectiveness, and item fit statistics for those 6 dimensions. The analysis shows that the final version of the 24-item ARP-CUS across 5 subscales for assessing students’ response to adversity, including control, attribution, reach, endurance, and transcendence. Overall, ARP-CUS demonstrates satisfactory psychometric properties for quantifying the adversity quotient of Chinese university students. While the study was properly conducted and the paper clearly written, I do have several minor questions for the authors' clarification.  The first one is about the sampling strategy. The authors did not say anything about how the sample was selected. What was the population from whom the sample was taken? Was it a convenient sample or a sample that was obtained through stratefied sampling?  Also of concern is the insufficient documentation of the context in which this study was conducted. Could the authors provide more information about this so that international readers will be better able to interpret the results?  More significantly, how does the study have implications for other samples in other contexts?

Author Response

Dear reviewer,

Thanks for your comments.

Best

WX

Reviewer 3 Report

Thank you for the opportunity to review the paper, "Re-developing the Adversity Response Profile for Chinese University Students".

The study is well conducted and the quality of the manuscript is satisfactory. Below are my comments for the authors to further improve the paper.

Introduction

  1. Lines 39-45, the authors indicated the problem statement of the study. However, more details are needed to help readers understand the two limitations. After reading the whole introduction, I found the authors presented a further illustration of the two limitations in lines 80 to 102. To enhance the clarity, the authors are suggested to (a) inform readers at the end of the first paragraph that further information will be provided in the following sections and (b) use subsection titles to guide readers read through the paper and to organize the contents.
  2. Line 69, the acronym of the Adversity Response Profile, (APR), is wrong.
  3. Lines 86-93, further explain the perspective of Confucianism and the philosophy of Taoism to help readers understand how both of them lead to the new AQ dimension. Similarly, the new dimension, transcendence, deserves further elaboration.    

Methodology

  1. Table 1, the total percentage of the year of study is 99.9. Please check.
  2. Lines 134-141, the writing is hard to follow. For instance, the meaning of "with the proportion of the right option in total options being more than 0.5" and "4 items directly translated from the original items" are not clear.
  3. I appreciate the authors' effort in providing the English version of the items. The authors are suggested to further explain the translation procedure and the qualification of the people who carried out the translation. Moreover, it is essential to remind readers that the English version was not validated in the study.
  4. Note that the eigenvalue and scree plot methods are not without limitations. It would be great to conduct parallel analysis or other tests to verify the number of factors to be retained in EFA.
  5. For CFA, use the robust maximum likelihood estimation unless the data are normally distributed.
  6. Justify the sample size or demonstrate the sample size has sufficient statistical power.

Results

  1. Show all factor loading values of the items in Table 2 and bold the highest values.
  2. Model 2 shows an acceptable but not good fit. The authors may want to consider and test other competing models (e.g., the single-factor model). Moreover, in Model 2, the Ownership and Origin dimensions were explained by a second-order factor (Attribution). However, as shown in Table 4, the correlation between Reach and Endurance dimensions is even stronger than the correlation between Ownership and Origin dimensions. Can the Reach and Endurance dimensions be explained by a second-order factor as well? It is also intriguing to know whether the Ownership and Origin dimensions can be combined.
  3. Present the results for the overall score of ARP-CUS in Table 4 as the dimensions are explained by a general AQ.

Discussion

  1. Lines 307-308, the non-significant relationship between transcendence and MF shall be discussed and explained. 

Author Response

(The authors gave the same response as above.)
